

**LES Modeling of Tsunami-like Solitary Wave Processes**
**over Fringing Reefs**
Yu Yao[1, 4], Tiancheng He[1], Zhengzhi Deng[2*], Long Chen[1, 3], Huiqun Guo[1]
[1] School of Hydraulic Engineering, Changsha University of Science and
Technology, Changsha, Hunan 410114, China.
[2] Ocean College, Zhejiang University, Zhoushan, Zhejiang 316021, China.
[3] Key Laboratory of Water-Sediment Sciences and Water Disaster Prevention of
Hunan Province, Changsha 410114, China.
[4] Key Laboratory of Coastal Disasters and Defence of Ministry of Education,
Nanjing, Jiangsu 210098, China
* Corresponding author: Zhengzhi Deng

17       E-mail: zzdeng@zju.edu.cn

18       Tel: +86 15068188376





**ABSTRACT**
Many low-lying tropical and sub-tropical reef-fringed coasts are vulnerable to
inundation during tsunami events. Hence accurate prediction of tsunami wave
transformation and runup over such reefs is a primary concern in the coastal management
of hazard mitigation.  To overcome the deficiencies of using depth-integrated models in
modeling tsunami-like solitary waves interacting with fringing reefs, a three-dimensional
(3D) numerical wave tank based on the Computational Fluid Dynamics (CFD) tool
OpenFOAM® is developed in this study. The Navier-Stokes equations for two-phase
incompressible flow are solved, using the Large Eddy Simulation (LES) method for
turbulence closure and the Volume of Fluid (VOF) method for tracking the free surface.
The adopted model is firstly validated by two existing laboratory experiments with
various wave conditions and reef configurations. The model is then applied to examine
the impacts of varying reef morphologies (fore-reef slope, back-reef slope, lagoon width,
reef-crest width) on the solitary wave runup. The current and vortex evolutions associated
with the breaking solitary wave around both the reef crest and the lagoon are also
addressed via the numerical simulations.

**Keywords:** Solitary wave; wave transformation, wave runup; fringing reef; LES.

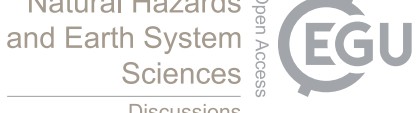

# 1 Introduction

Tsunami is an extremely destructive natural disaster, which can be generated by earthquakes, landslides, volcanic eruptions, and meteorite impacts. Tsunami damage occurs mostly in the coastal areas where tsunami waves runup or rundown the beach, overtop or ruin the coastal structures, and inundate the coastal towns and villages (Yao et al., 2015). Some tropic and sub-tropic coastal areas vulnerable to tsunami hazards are surrounded by coral reefs, especially those in the Pacific and Indian Oceans. Among various coral reefs, fringing reefs are the most common type. A typical cross-shore fringing reef profile can be characterized by a steep offshore fore-reef slope and an inshore shallow reef flat (Gourlay, 1996). There is also possibly a reef crest lying at the reef edge (e.g., Hench et al., 2008) and/or a narrow shallow lagoon existing behind the reef flat (e.g., Lowe et al., 2009a). Over decades, fringing reefs have been well recognized to be able to shelter low-lying coastal areas from flood hazards associated with storms and high surf events (e.g. Cheriton et al. 2016; Lowe et al., 2005; Lugo-Fernandez et al., 1998; Péquignet et al., 2011; Young, 1989). However, until after the 2004 Indian Ocean Tsunami, the positive role of coral reefs in mitigating the tsunami waves has begun to arise the attentions of the scholars who conducted the post-disaster surveys (e.g., Chatenoux and Peduzzi, 2007; Ford et al., 2014; Mcadoo et al., 2011). There is consensus among the scholars that in addition to establish the global tsunami warning system, the cultivation of coastal vegetation (mangrove forest, coral reef, etc.) is also one of the coastal defensive measures against the tsunami waves (e.g., Dahdouh-Guebas et al., 2006; Danielsen et al., 2005; Mcadoo et al., 2011). Numerical models have been proven to be powerful tools to investigate tsunami wave interaction with the mangrove forests (e.g., Huang et al., 2011; Maza et al., 2015; Tang et al., 2013 and many others). Comparatively speaking, their applications in modeling coral reefs subjected to tsunami waves are still very few.

Over decades, modeling wave processes over reef profiles faces several challenges such as steep fore-reef slope, complex reef morphology as well as spatially-varied surface roughness. Local but strong turbulence due to wave breaking in the vicinity of reef edge needs to be resolved. Among various approaches for modelling wave dynamics over reefs, two groups of models are the most pervasive. The first group focuses on using the





phase-averaged 2DH and 3D models to study both the wind waves and the mean flows in
field reef environments, and typically the radiation stress concept (Longuet-Higgins and
Stewart, 1964) is used to couple the waves and the flows (e.g., Douillet et al., 2001;
Kraines et al., 1998; Lowe et al., 2009b, 2010; Van Dongeren et al., 2013; Quataert et al.,
2015). As for modeling tsunami waves at a field scale, we are only aware of in the
literature that Kunkel et al. (2006) implemented a nonlinear shallow water model to study
the effects of wave forcing and reef morphology variations on the wave runup. However,
their numerical model was not verified by any field observations. The second group aims
at using the computationally efficient and phased-resolving model based on the
Boussinesq equations. This depth-integrated modeling approach employs a polynomial
approximation to the vertical profile of velocity field, thereby reducing the dimensions of
a three-dimensional problem by one. It is able to account for both nonlinear and
dispersive effects at intermediate water level. At a laboratory scale, Boussinesq models
combined with some semi-empirical breaking-wave and bottom friction models have
been proven to be able to simulate the motions of regular waves (Skotner and Apelt, 1999;
Yao et al., 2012), irregular waves (Nwogu and Demirbilek, 2010; Yao et al., 2016, 2019)
and infragravity waves (Su et al., 2015; Su and Ma, 2018) over fringing reef profiles.

The solitary wave has been employed in many laboratory/numerical studies to model

the leading wave of a tsunami. Compared to the aforementioned regular/irregular waves,
the numerical investigations of solitary wave interaction with the laboratory reef profile
are much fewer. Roeber and Cheung (2012) was the pioneer study to simulate the solitary
wave transformation over a fringing reef using a Boussinesq model. Laboratory
measurements of the cross-shore wave height and current across the reef as conducted by
Roeber (2010) were reproduced by their model. More recently, Yao et al. (2018) also
validated a Boussinesq model based on their laboratory experiments to assess the impacts
of reef morphologic variations (fore-reef slope, back-reef slope, reef-flat width, reef-crest
width) on the solitary wave runup over the back-reef beach. Despite of above applications,
several disadvantages still exist in using the Boussinesq-typed models: (1) Boussinesq
equations are subjected to the mild-slope assumption, thus it is questionable when using
for reefs with steep fore-reef slope, particularly when there is a sharp reef crest locating at
the reef edge; (2) wave breaking could not be inherently captured by Boussinesq-type





models thus empirical breaking model or special numerical treatment is usually needed;
(3) Boussinesq models could not resolve the vertical flow structure associated with the
breaking waves due to the polynomial approximation to the vertical velocity profile.
To remedy the above deficiencies of using Boussinesq-typed models to simulate the
solitary processes (wave breaking, bore propagation, and runup) over the fringing reefs,
we develop a 3D numerical wave tank based on the CFD tool OpenFOAM® (Open Field
Operation and Manipulation) in this study. OpenFOAM® is a widely used open-source
CFD code in the modern industry supporting two-phase incompressible flow (via its
solver interFoam). With appropriate treatment of wave generation and absorption, it has
been proved to be a powerful and efficient tool for exploring complicated nearshore wave
dynamics (e.g., Higuera et al., 2013b). In this study, the Navier–Stokes equations for an
incompressible fluid are solved. For the turbulence closure model, although LES
demands more computational resources than RANS, it computes the large-scale unsteady
motions explicitly. Importantly, it could provide more statistical information for the
turbulence flows in which large-scale unsteadiness is significant (Pope, 2000). Thus the
LES model is adopted by considering that the breaking-wave driven flow around the reef
edge/crest is fast and highly unsteady. The free surface motions are tracked by the widely
used VOF method.
In this study, we first validate the adopted model by the laboratory experiments of
Roeber (2010) as well as our previous experiments (Yao et al., 2018). The robustness of
the present model in reproducing such solitary wave processes as wave breaking near the
reef edge/crest, turbulence bore propagating on the reef flat and wave runup on the back-
reef beach, is demonstrated. The model is then applied to investigate the impacts of
varying reef morphologies (fore-reef slope, back-reef slope, lagoon width, reef crest
width) on the solitary wave runup. The flow and vorticity fields associated with the
breaking solitary wave around the reef crest and the lagoon are also analyzed by the
model results. The rest of this paper is organized as follows. The numerical model is
firstly described in Section 2. It is then validated by the laboratory data from the literature
as well as our data in Section 3. What follows in Section 4 are the model applications for
which laboratory data are unavailable. The main conclusions drawn from this study are
given in Section 5.





## 2 Numerical Methods

### 2.1 Governing equations



To simulate breaking-wave processes across the reef, the LES approach is employed

to balance the need of resolving a large portion of the turbulent flow energy in the
domain while parameterizing the unresolved field with a subgrid closure in order to
maintain a reasonable computational cost. The filtered Navier-Stokes equations is
essential to separate the velocity field that contains the large-scale components, which is
performed by filtering the velocity field (Leonard, 1975). The filtered velocity is defined
as
$$\bar{u}_i(x) = \int G(x, x') u_i(x') dx' \tag{1}$$

where $G(x, x')$ is the filter kernel, which is a localized function. The eddy sizes are
identified using a characteristic length scale, $\Delta$, which is defined as
$$\Delta = (\Delta x \cdot \Delta y \cdot \Delta z)^{1/3} \tag{2}$$

where $\Delta x$, $\Delta y$, $\Delta z$ are the grid size in streamlines, spanwise and vertical directions,
respectively. Eddies that are larger than $\Delta$ are roughly considered as large eddies, and
they are directly solved. Those who are smaller than $\Delta$ are small eddies.

The filtered continuity and momentum equations are as follows

$$\frac{\partial \bar{u}_i}{\partial x_i} = 0 \tag{3}$$

$$\frac{\partial \rho \bar{u}_i}{\partial t} + \frac{\partial (\rho \bar{u}_i \bar{u}_j)}{\partial x_i} = -\frac{\partial \bar{p}}{\partial x_i} + \rho g_i + 2\mu \frac{\partial \bar{S}_{ij}}{\partial x_j} - \frac{\partial \tau_{ij}^r}{\partial x_j} \tag{4}$$

where $\bar{p}$ is the filtered pressure, $\bar{S}_{ij}$ is the strain rate of the large scales defined as
$$\bar{S}_{ij} = \frac{1}{2}\left(\frac{\partial \bar{u}_i}{\partial x_j} + \frac{\partial \bar{u}_j}{\partial x_i}\right) \tag{5}$$

and $\tau_{ij}^r$ is the residual stress approximated by using sub-grid scale (SGS) models to get a
full solution for the Navier-Stokes equations.

The SGS stress is usually calculated by a linear relationship with the rate of strain

tensor based on the Boussinesq hypothesis. The one-equation eddy viscosity mode, which



is supposed to be better than the well-known Smagorinsky model for solving the highly
complex flow and shear flow (Menon et al., 1996), is employed in the present study.
Based on the one-equation model (Yoshizawa and Horiuti, 1985), the sub-grid stresses
are defined as
$$\tau_{ij}^r = \frac{2}{3}k_s\delta_{ij} - 2v_t(\overline{S}_{ij} - \frac{1}{3}\overline{S}_{kk}\delta_{ij}) \tag{6}$$

where $\delta_{ij}$ is the Kronecker-delta, and $v_t$ is the SGS eddy viscosity, which is given by
$$v_t = C_k\overline{\Delta}\sqrt{k_s} \tag{7}$$

and the SGS kinetic energy $k_s$ needs to be solved by
$$\frac{\partial k_s}{\partial t} + \overline{u}_i\frac{\partial k_s}{\partial x_i} = \frac{\partial}{\partial x_i}(\frac{\mu}{P_r}\frac{\partial k_s}{\partial x_i}) - \frac{\tau_{ij}^r}{\rho}\frac{\partial \overline{u}_j}{\partial x_i} - \frac{C_\varepsilon k_s^{3/2}}{\Delta} \tag{8}$$

where $C_k = 0.094$, $C_\varepsilon = 0.916$ and $P_r = 0.9$ as suggested by the OpenFOAM® User
Guide (2013).

The fluid field in the present study consists of water and air, and both phases are

solved using the VOF method (Hirt and Nichols, 1981). The general representation of
fluid density $\rho$ is written as
$$\rho = \alpha\rho_1 + (1-\alpha)\rho_2 \tag{9}$$

where $\rho_1 = 1000\,kg/m^3$ is the density of water, $\rho_2 = 1\,kg/m^3$ is the density of air, $\alpha$ is
the volume fraction of water contained in a grid cell. The distribution of $\alpha$ is modeled by
an advection equation
$$\frac{\partial \alpha}{\partial t} + \nabla \cdot (\alpha\overline{u}_i) + \nabla \cdot [\alpha(1-\alpha)u_i^r] = 0 \tag{10}$$

The last term on the left side is an artificial compression term, avoiding the excessive
numerical diffusion and the interface smearing, the new introduced $u_i^r$ is a velocity field
suitable to compress the interface.

In the present solver interFoam, the algorithm PIMPLE, which is a mixture of the

PISO (Pressure Implicit with Splitting of Operators) and SIMPLE (Semi-Implicit Method
for Pressure-Linked Equations) algorithms, is employed to solve the coupling of velocity
and pressure fields. The MULES (multi-dimensional universal limiter for explicit



solution) method is used to maintain boundedness of the volume fraction independent of
the underlying numerical scheme, mesh structure, *etc*. Euler scheme is utilized for the
time derivatives, Gauss linear scheme is used for gradient term, and Gauss linear
corrected scheme is selected for the Laplacian term. Detailed implementation can be
founded in the OpenFOAM® User Guide (2013).
**2.2 Wave generation and absorption**
Wave generation and absorption are essentials for a numerical wave tank, but they
are not included in the official version of OpenFOAM®. Therefore, supplementary
modules were developed by the other users, e.g., waves2Foam (Jacobsen et al., 2012) and
IH-FOAM (Higuera et al., 2013a). In this study, the IH-FOAM is selected in that it
employs an active wave absorbing boundary and does not require an additional relaxation
zone as used by waves2Foam. Meanwhile, it supports many wave theories including the
solitary wave theory. The free surface and velocity for a solitary wave generation in IH-
FOAM are (Lee et al., 1982)

$$\eta = H \operatorname{sech}^2\left( \sqrt{\frac{3H}{4h^3}} X \right) \tag{11}$$

$$\frac{u}{\sqrt{gh}} = \frac{\eta}{h}\left[ 1 - \frac{1}{4}\frac{\eta}{h} + \frac{h}{3}\frac{h}{\eta}\left( 1 - \frac{3}{2}\frac{z^2}{h^2} \right)\frac{d^2\eta}{dX^2} \right] \tag{12}$$

$$\frac{w}{\sqrt{gh}} = \frac{-z}{h}\left[ \left( 1 - \frac{1}{2}\frac{\eta}{h} \right)\frac{d\eta}{dX} + \frac{1}{3}h^2\left( 1 - \frac{1}{2}\frac{z^2}{h^2} \right)\frac{d^3\eta}{dX^3} \right] \tag{13}$$

where $\eta$ is the free surface elevation, $H$ is the wave height, $h$ is the water depth,
$X = x - ct$, $c = \sqrt{g(h+H)}$ is the wave celerity, $u$ and $w$ are the velocities in the
streamwise and vertical directions, respectively.
**3 Model validation**
**3.1 Experimental settings**
The first set of laboratory experiments serving as validation purpose is Roeber (2010),
who reported two series of experiments conducted at Oregon State University, U.S.A. in

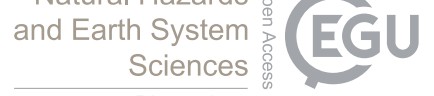

separate wave flumes. In this study, we only reproduce their experiments in the large
wave flume, which is 104 m long, 3.66 wide and 4.57 m high. As illustrated in Fig. 1a,
the two-dimensional (2D) reef model, starting at 25.9 m from the wavemaker, was built
by a plane fore-reef slope attached to a horizontal reef flat of 2.36 m high followed by a
back-reef vertical wall. Both the waves and flows across the reef profile were measured
by 14 wave gauges (wg1-wg14) and 5 ADVs (Acoustic Doppler velocimeters),
respectively. Only two scenarios for the reef with and without a trapezoidal reef crest
subjected to two incident waves are reported in this study (see also Table 1). The large
wave flume experiments facilitate us to test our model's ability to handle relatively large-
scale nonlinear dispersive waves together with wave breaking, bore propagation and
associated wave-driven flows. For more detailed experimental setup, see Roeber (2010).

The second set of 2D reef experiments for model validation comes from our

previous work (Yao et al., 2018). These experiments were conducted in a small wave
flume 40 m long, 0.5 m wide and 0.8 m high at Changsha University of Science and
Technology, P. R. China. As shown in Fig. 1b, a plane slope was built at 27.3 m from the
wavemaker and it was truncated by a horizontal reef flat of 0.35 m high. A back-reef
beach of 1:6 was attached to the end of the reef flat. The surface elevations were
measured at 8 cross-shore locations (G1-G8) and no flow measurement was performed.
However, A CCD camera was installed to record the process of water uprush on the
back-reef slope. Thus the model's robustness to capture the whole process of solitary
wave transformation over the reef flat and runup on the back-reef beach can be evaluated.
In this study, we only simulate the tested idealized reef profile with and without a lagoon
at the rear of reef flat subjected to the same wave condition (see also Table 1). The
lagoon was formed by two 1:1 slope connecting the reef flat and the toe of the back-reef
beach to the flume bottom, respectively. See Yao et al. (2018) for the detailed laboratory
settings.


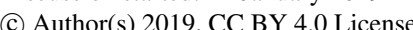

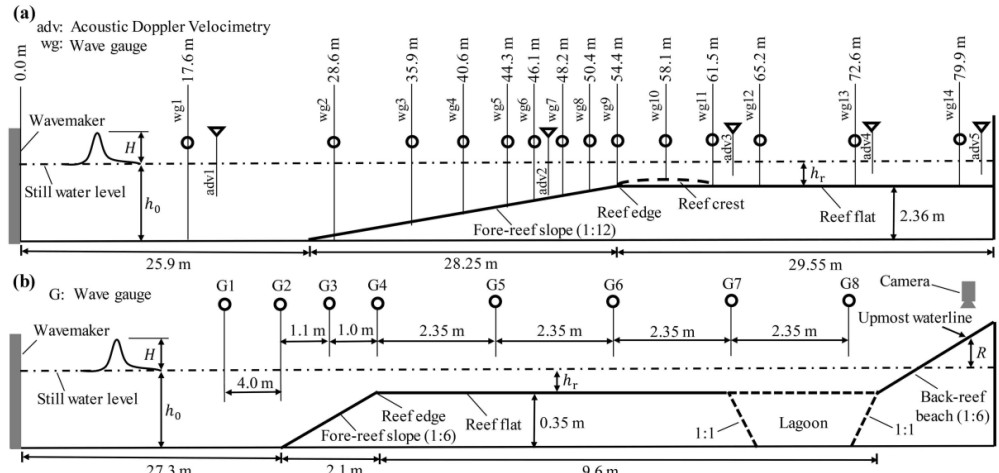


Fig. 1 Experiment settings for: (a) Roeber (2010) and (b) Yao et al. (2018).

Table 1 Reef configuration and wave condition for the tested scenarios

| Scenario I.D. | Offshore wave height $H_0$ (m) | Offshore water depth $h_0$ (m) | Reef-flat water depth $h_r$ (m) | Fore-reef slope $s$ | Reef-flat length $L_r$ (m) | Remarks | Source |
|---|---|---|---|---|---|---|---|
| 1 | 1.23 | 2.46 | 0.1 | 1:12 | 29.5 | – | Roeber (2010) |
| 2 | 0.75 | 2.5 | 0.14 | 1:12 | 22.8 | With reef crest | Roeber (2010) |
| 3 | 0.08 | 0.40 | 0.05 | 1:6 | 9.6 | – | Yao et al. (2018) |
| 4 | 0.08 | 0.40 | 0.05 | 1:6 | 8.0 | With lagoon | Yao et al. (2018) |


**3.2 Numerical settings**

By considering a balance between the computational accuracy and efficiency, the

computational domain (Fig. 2a) is designed to reproduce the main aspects of the
laboratory settings. We calibrate the model in the principle that the computed leading
solitary wave height at the most offshore gauge should exactly reproduce its
measurement. For a solitary wave, wave length ( $L$ ) can be estimated as a distance
containing 95% of the total mass of the solitary wave, which yields $L = 2.12h / \sqrt{H_i / h}$ .




The largest offshore wave length according to the wave conditions in Table 1 is $L$ =8.44
m/1.52 m for the scenario of Roeber (2010)/Yao et al. (2018). Thus, we reasonably put
the numerical wave generation and absorption at a location 15 m/6 m from the toe of
fore-reef slope, which is also the location of left boundary. Behind the reef flat,
transmitted waves are allowed to runup on the back-reef beach, but they cannot overtop
out of the computational domain due to a solid wall condition at the right boundary. In
addition, we set the "free to the atmosphere" for the top boundary and the "no-slip wall"
condition at the bottom. For the two side faces, we employed the "empty" boundary in
OpenFOAM to simulate the 2D reef configurations.
Structured mesh is used to discretize the computational domain. The discretization is
kept constant in spanwise ( $y$ ) direction (one layer of 20 mm/10 mm for Roeber/Yao et
al.'s scenarios) and vertical ( $z$ ) direction (20 mm/8 mm for Roeber/Yao et al.'s
scenarios), while it varies in the streamwise ( $x$ ) direction to reduce the number of the
total cells. From the left boundary to the toe of the fore-reef slopes, $\Delta x$ decreases
gradually from 100 mm/24 mm to 20 mm/8 mm for Roeber/Yao et al.'s scenarios (see
e.g., Figs. 2b and 2c). The core region (see e.g., Fig. 2d), covering from the fore-reef
slope to the back-reef wall or beach, maintains a constant cell size of $\Delta x$= 20 mm and 8
mm for the two experiments, respectively. Grid refinement near the free surface (e.g.,
Figs. 2b and 2c) is conducted across the domain in both $x$ and $z$ directions by reducing
the grid sizes to one-quarter of their original values, e.g., $\Delta x$= 5 mm/2 mm and $\Delta z$= 5
mm/2 mm at the core region. The total computational mesh consists of 4.87 million/1.18
million cells for Roeber/Yao et al.'s scenarios. The simulation duration is appointed to be
80 sec/30 sec to guarantee the arrival of the reflected waves at the most offshore wave
gauge in both experiments. The time step is automatically adjusted during computation
for a constant Courant number of 0.25. Via parallel computing, it takes approximately
16d /2d for Roeber/Yao et al.'s scenarios on a cluster server with 44 CPUs (Intel Xeon,
E5-2696, 2.2 G). No notable improvement of the results could be found with further
refinement of the grid size.

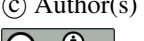




Fig. 2 Numerical grids and boundary conditions of the numerical domain.

To evaluate the performance of the model, the model skill value is adopted and

calculated by Wilmott (1981)

$$skill = 1 - \frac{\sum \left| Y_{model} - Y_{obs} \right|^2}{\sum \left( \left| Y_{model} - \overline{Y_{obs}} \right| + \left| Y_{obs} - \overline{Y_{obs}} \right| \right)^2} \qquad (14)$$


where $Y_{model}$ is the predicted value, $Y_{obs}$ is the measured value. The upper dash indicates


that the average value is taken. The higher the skill number (close to 1), the better


performance of the numerical model.

**3.3 Comparison between numerical and experimental results**

Fig. 3 compares the computed and the measured cross-shore distribution of the free

surface elevations ($\eta$) at different stages ($t$) for Scenario 1, where $\eta$ is normalized by
the offshore still water depth ($h_0$) and $t$ is normalized by $\sqrt{h_0 / g}$. Incident solitary wave
gets steepened on the fore-reef slope at $t / \sqrt{h_0 / g} = 62.3$ due to the shoaling effect. Then
its front becomes vertical prior to breaking at $t / \sqrt{h_0 / g} = 64.3$. At $t / \sqrt{h_0 / g} = 65.8$, a
plunging breaker occurs with air entrainment and splash-up near the reef edge. After that,
breaking wave starts to travel on the reef flat in the form of a propagating turbulent bore
at $t / \sqrt{h_0 / g} = 67.1$. The bore shows a gradual reduction in amplitude and continues to
propagate downstream on the reef flat at $t / \sqrt{h_0 / g} = 76.3$. The numerical results
generally agree well with the laboratory measurements at all stages with the skill values
larger than 0.85, indicating the robustness of the adopted model to address the solitary
wave processes across the laboratory reef profile in the large wave flume. When
comparing the predictions between our Navier-Stokes-equation-based model and a





Boussinesq model adopted by Roeber (2010), it seems that our model better captures the
steep near breaking wave ( $t / \sqrt{h_0 / g} = 64.3$ ) and breaking wave ( $t / \sqrt{h_0 / g} = 65.8$ ).

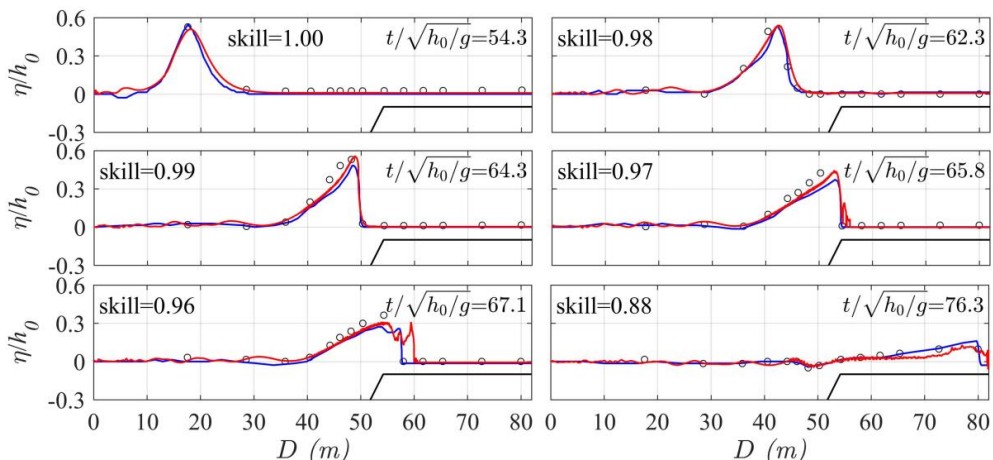


Fig. 3 Dimensionless free surface elevations ( $\eta / h_0$ ) across the reef at different stages
( $t / \sqrt{h_0 / g}$ ) for Scenario 1. Red lines - present simulations; Blue lines - simulations from
Roeber (2010); Open circles - measurements from Roeber (2010); Skill values are for the
present simulations.

Fig. 4 illustrates the computed and measured time-series of dimensionless free
surface elevations ( $\eta / h_0$ ) at different cross-shore locations ( $D$ ) for Scenario 1. It
appears that the model reasonably simulates the transformation processes of solitary wave
on the fore-reef slope ( $D = 35.9 \, \text{m}$ and $44.3 \, \text{m}$ ) and near the reef edge ( $D = 50.4 \, \text{m}$ )
with the skill values larger than 0.9. The skill values become relatively lower right after
the incipient wave breaking point ( $D = 57.9 \, \text{m}$ ) and at the central reef flat ( $D = 65.2 \, \text{m}$ ).
Such discrepancies may be primarily due to the air entrainment in measuring both the
breaking wave and the moving bore. In addition, the second peaks in the time series are
due to wave reflection from the back-reef wall, which are well predicted by the present
model. Meanwhile, no notable difference could be found in view of the time-series





predictions between the present model and the model of Roeber (2010), except at
$D = 65.2$ m where the bore amplitude decays in our simulation compared to that at
$D = 57.9$ m .

Fig. 4 Time-series of dimensionless free surface elevations ($\eta / h_0$) at different cross-


shore distances from the wavemaker ($D$) for Scenario 1. Red lines - present simulations;
Blue lines - simulations from Roeber (2010); Open circles - measurements from Roeber
(2010); Skill values are for the present simulations.

Fig. 5 depicts the time-series of streamwise velocity ($u$) at five cross-shore
locations ($D$) for Scenario 1, in which $u$ is normalized by the local shallow water wave
speed ($\sqrt{gh}$). The model satisfactorily captures the measured velocity offshore
($D = 17.8$ m ), on the fore-reef slope ($D = 47.4$ m ), on the central reef flat ($D = 72.6$ m )
and near the shoreline ($D = 80.2$ m ). A transition from the subcritical flow ($u / \sqrt{gh} < 1$)
to supercritical flow ($u / \sqrt{gh} > 1$) could be observed right after wave breaking
($D = 61.6$ m ), and less satisfactory prediction (skill values =0.76) at this location is

probably again due to the effect of air-bubbles during the flow measurements. Overall,


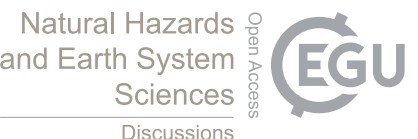

the adopted model outperforms the Boussinesq model of Roeber (2010) in view of the
velocity predictions, particularly both near the breaking point ( $D = 61.6\,\mathrm{m}$ ) and the
shoreline on the reef flat ( $D = 80.2\,\mathrm{m}$ ).

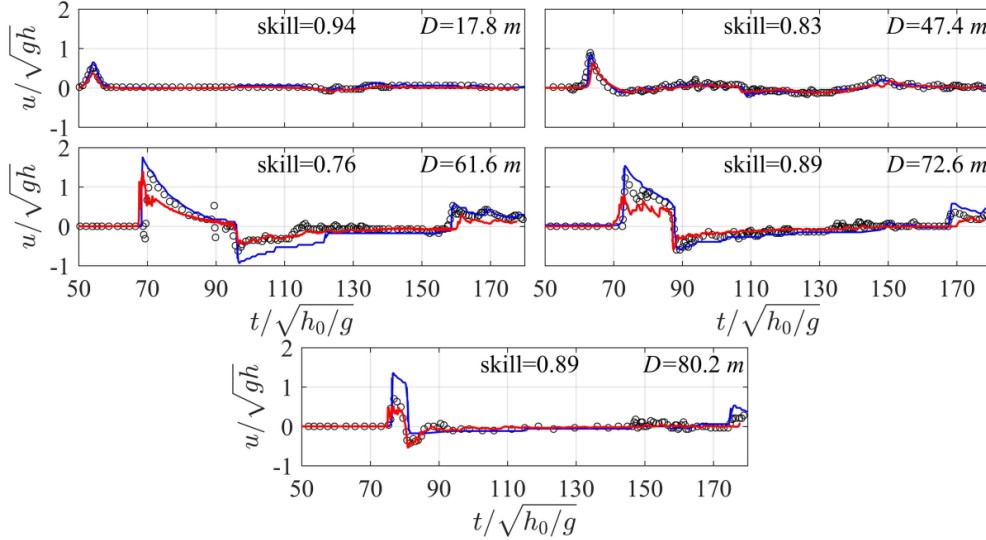

Fig. 5 Time-series of dimensionless streamwise velocity ( $u / \sqrt{gh}$ ) at different cross-
shore distances from the wavemaker ( $D$ ) for Scenario 1. Red lines - present simulations;
Blue lines - simulations from Roeber (2010); Open circles - measurements from Roeber
(2010); Skill values are for the present simulations.

As previously introduced, the reef profile of Scenario 2 is identical to that of

Scenario 1 except for a reef crest locating at the reef edge. The cross-shore distribution of
dimensionless free surface elevations ( $\eta / h_0$ ) at different stages ( $t / \sqrt{h_0 / g}$ ) for Scenario
2 is demonstrated in Fig. 6. Steepened shoaling wave on the fore-reef slope appears at
$t / \sqrt{h_0 / g} = 65.0$ and its front becomes almost vertical prior to breaking at
$t / \sqrt{h_0 / g} = 66.5$ . Breaking wave begins to overtop over the reef crest
( $t / \sqrt{h_0 / g} = 69.1$ ), and it then collapses on the leeside of reef crest, resulting in a moving



turbulent bore ($t / \sqrt{h_0 / g} = 72.5$). The bore travels shoreward on the reef flat with the
continuous damping of its magnitude ($t / \sqrt{h_0 / g} = 80.5$). The skill values for all
sampling locations in this Scenario are larger than 0.9, implying that the adopted model is
able to well address the solitary wave processes over a more complicated reef geometry
such as the presence of a reef crest at the reef edge. Again, the present model predicts the
near breaking wave (` $t / \sqrt{h_0 / g} = 66.5$) and breaking wave ($t / \sqrt{h_0 / g} = 69.1$ and
$t / \sqrt{h_0 / g} = 72.5$) slightly better than the model adopted by Roeber (2010).

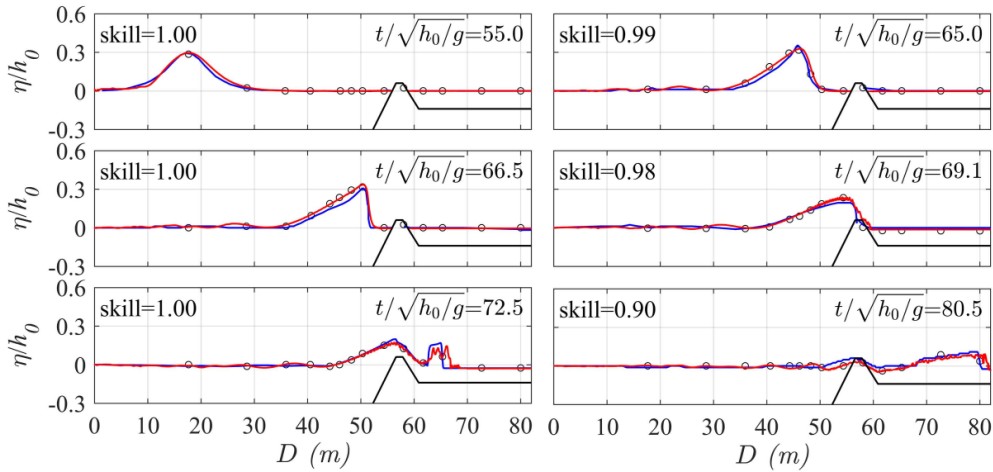


Fig. 6 Dimensionless free surface elevations ($\eta / h_0$) across the reef at different stages
($t / \sqrt{h_0 / g}$) for Scenario 2. Red lines - present simulations; Blue lines - simulations from
Roeber (2010); Open circles - measurements from Roeber (2010); Skill values are for the
present simulations.

Fig. 7 compares the measured and simulated times-series of dimensionless free

surface elevations ($\eta / h_0$) at various cross-shore locations ($D$) for Scenario 2. The skill
values at all locations are larger than 0.85. It suggest again that the present model not
only reasonably reproduces wave propagation offshore ($D = 17.6$ m), shoaling on the



fore-reef slope ( $D = 35.9$ m and $44.3$ m ) and near breaking in front of the reef crest
( $D = 50.4$ m ), breaking-wave transformation over the reef crest ( $D = 57.9$ m ), and bore
evolution on the reef flat ( $D = 65.2$ m ), but also captures the tail waves caused by wave
reflection from the back-reef wall ( see e.g., $D = 65.2$ m ). Overall, both our model and
the model of Roeber (2010) reproduce the timeseries of free surface elevations equally
well for this scenario.

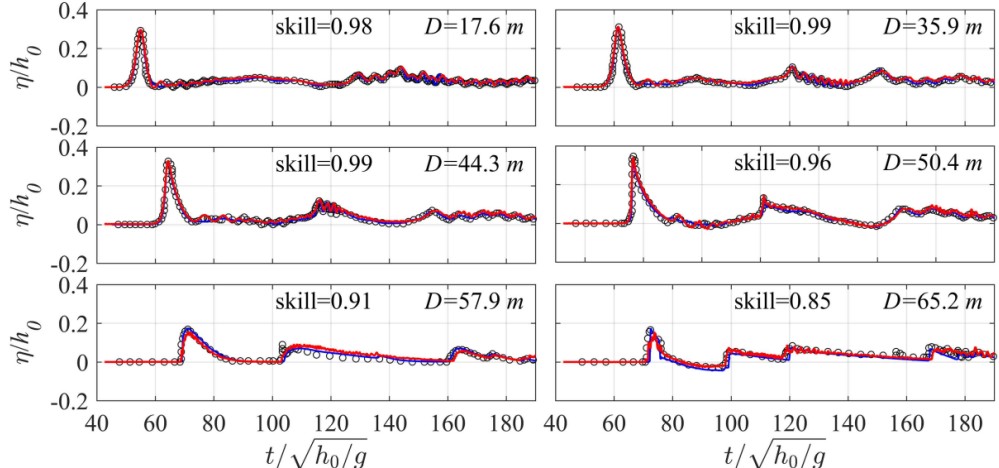


Fig. 7 Time-series of dimensionless free surface elevations ( $\eta / h_0$ ) at different cross-
shore distances from the wavemaker ( $D$ ) for Scenario 2. Red lines - present simulations;
Blue lines - simulations from Roeber (2010); Open circles - measurements from Roeber
(2010); Skill values are for the present simulations.

As for Scenario 2, Roeber (2010) only reported one location of flow measurement

on the seaside face of the reef crest. Fig. 8 presents the time-series of dimensionless
streamwise velocity ( $u / \sqrt{gh}$ ) at the point ( $x = 54.4$ m ), and a skewed and peaky
velocity profile is observed associated with the leading solitary wave because the position
is very close to the incipient wave breaking point. The two secondary peaks in the time
series are generated by the reflected waves from the reef crest and from the back-reef
wall, respectively. The model captures the temporal variation of current fairly well with



the skill value of 0.86, and its prediction is also better than that from the model of Roeber
(2010), particularly for the reflected waves.

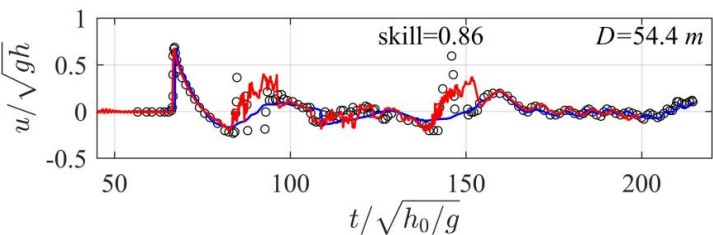

Fig. 8 Time-series of dimensionless streamwise velocity ($u/\sqrt{gh}$) at the cross-shore
distance $D = 54.4$ m from the wavemaker for Scenario 2. Red lines - present
simulations; Blue lines - simulations from Roeber (2010); Open circles - measurements
from Roeber (2010); Skill values are for the present simulations.

The experiments of Yao et al. (2018) only measured the timeseries of wave records
at limited locations (G1-G8) across the reef as well as the maximum wave runup on the
final beach. Fig. 9 compares the computed and measured time-series free surface
elevations for Scenario 3. The overall agreement between the simulations and
experiments for G1-G8 is very good with the skill values at all locations larger than 0.9.
When the solitary wave travels from the toe (G2) to the middle of fore-reef slope (G3), it
gets steepened due to the shoaling effect. Wave breaking starts at a location right before
the reef edge (G4) and the surfzone processes extend over the reef flat in the form of a
moving bore. Thus from G5 to G8, the wave timeseries show saw-shaped profiles and
there is a cross-shore decrease of the leading solitary wave height. Such features of the
breaking waves are also well captured by the model. Note that the second peak in the
timeseries of G7 is due to wave reflection from the back-reef beach, and the incident and
reflected waves are not fully separated from each other at G8 because this location is too
close to the beach. The predicted and measured wave runups are 0.122 m and 0.109 m,

respectively, for this scenario. Compared to the Boussinesq model employed by Yao et al.



(2018), no significant difference in the predicted timeseries could be found for the present
Navier-Stokes-equation-based model.

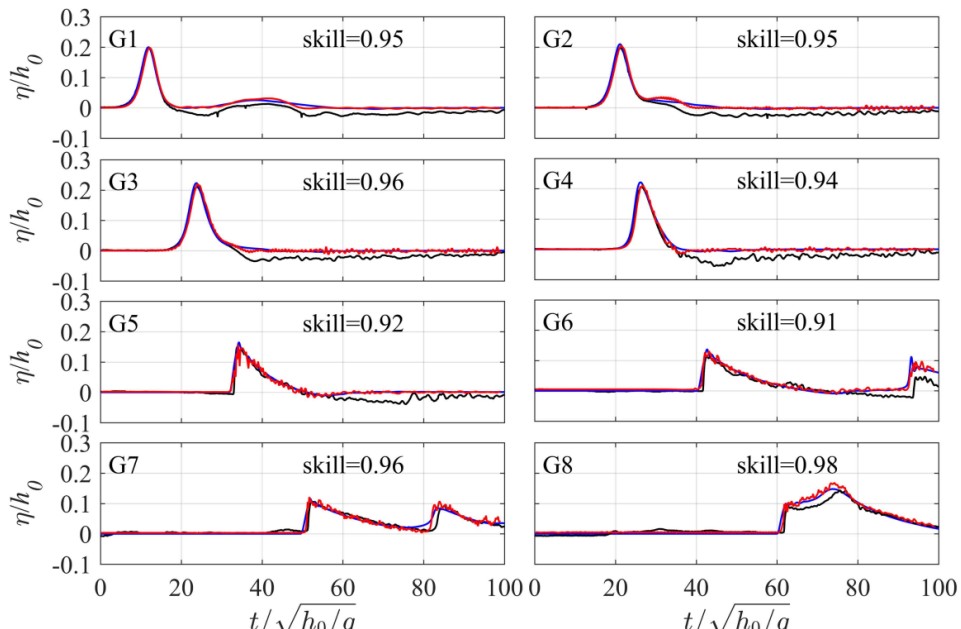


Fig. 9 Time-series of dimensionless free surface elevations ($\eta / h_0$) at different cross-
shore sampling locations (G1-G8) for Scenario 3. Red lines - present simulations; Blue
lines - simulations from Yao et al. (2008); Black lines - measurements from Yao et al.
(2008); Skill values are for the present simulations.

Fig. 10 depicts the same comparison of wave time-series but for the reef profile with
a lagoon (Scenario 4). Again, the model performance for this scenario is fairly good (all
skill values larger than 0.9). The predicted and measured wave runups are 0.123 m and
0.116 m, respectively, for this scenario. Notable mismatch only appears for those small
wave oscillations generated by the reflected wave propagating out of the lagoon to the
reef flat (i.e., from G8 to G6). But our model seems to be superior to the model of Yao et
al. (2018) to reproduce those oscillations at G7 and G8. We finally remark that the tail of

       leading solitary wave, particularly from G1 to G4, is below the initial water level in the



laboratory data, which is due to the water lost to form the generated wave crest around
the paddle of the wave maker. However, such phenomenon is not observed in the
numerical results because we generate a theoretical solitary wave in the numerical
domain as indicated by Eq. (11).

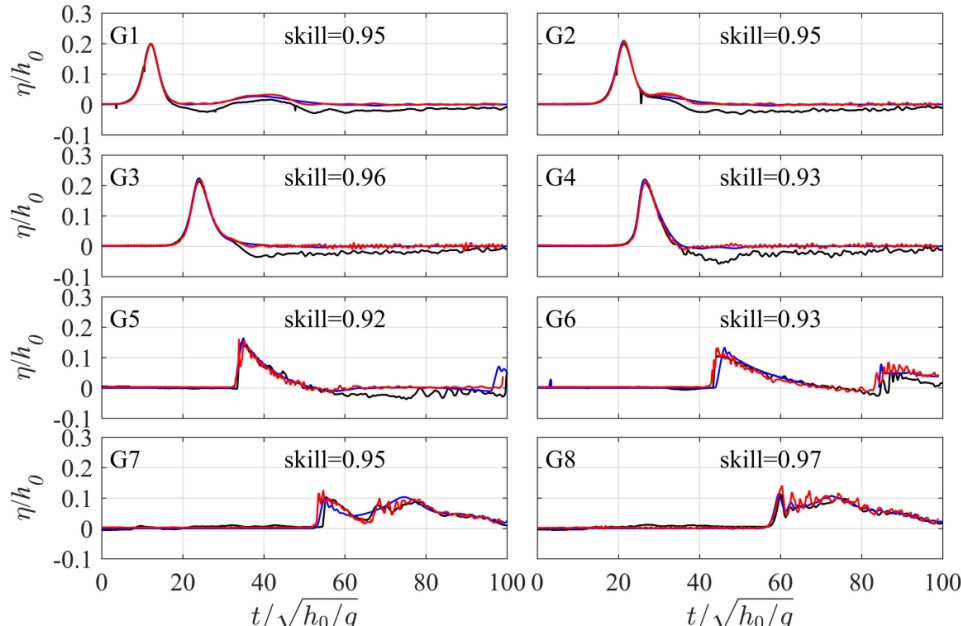


Fig. 10 Time-series of dimensionless free surface elevations ($\eta / h_0$) at different cross-
shore sampling locations (G1-G8) for Scenario 4. Red lines - present simulations; Blue
lines - simulations from Yao et al. (2008); Black lines - measurements from Yao et al.
(2008); Skill values are for the present simulations.
**4. Model Applications**
**4.1 Effects of reef morphology variations on the solitary wave runup**
In this section, we apply the well-validated LES model to examine the variations of
reef morphological parameters (fore-reef slope, back-reef slope, lagoon width, reef-crest
width) that may affect the wave runup ($R$) on the back-reef beach. Based on Scenario 3
(1: 6 for both the slopes of fore-reef and back-reef, 9.6 m for the reef length, no reef crest
and no lagoon) from Yao et al. (2018), we firstly test five slopes (1:2, 1:4, 1:6, 1:8 and



1:10, which all fall within the common range of 1:1 to 1:20 in the reported field
observations, see e.g., Quataert et al. 2015, their Table 1) for both the fore-reef and the
back-reef. We then consider the existence of a lagoon at the rear of reef flat by testing
four upper widths of the lagoon (1.6 m, 3.2 m, 4.8 m and 6.4 m) and comparing to the
case without lagoon (lagoon width=0 m). We finally investigate a trapezoidal reef crest
locating at the reef edge with its seaward slope matching the fore-reef slope and its
shoreward slope of 1:1. We examine five reef-crest widths (0.1 m, 0.2 m, 0.3 m, 0.4 m
and 0.5 m) in view that the dimension of reef crest at the field scale is on the magnitude
of meters (see e.g., Hench et al., 2008). During simulations, each run is performed by
changing one of above morphological parameters while keeping other parameters
unaltered. All runs are conducted under a combination of one solitary wave height
( $H_0 = 0.08 \, \mathrm{m}$ ) and two reef-flat water depths ( $h_r = 0.05 \, \mathrm{m}$ and $h_r = 0.1 \, \mathrm{m}$ ).

Generally, Fig. 11a shows that $R$ is not very sensitive to the change of the fore-reef

slope within the tested range, in that wave breaking for this scenario occurs closely to the
reef edge (G4), thus most of the surfzone processes and associated energy dispassion
complete on the reef flat. Only when the fore-reef slope becomes steeper than 1:8, $R$
decreases slightly under both water depths ( $h_r$ ), which is attributed to the increased fore-
reef reflection of the incident wave energy. Fig. 11b reveals that $R$ is more sensitive to
the back-beach slope under both $h_r$ . It decreases significantly with the growth of back-
reef beach slope, which is consistent with that found for the plane slope (see e.g.,
Synolakis, 1987). Fig. 11c shows the variation of $R$ with the lagoon width. Note that the
zero width represents the reef without lagoon.  It appears that $R$ increases notably with
the increase of lagoon width because a wider lagoon dissipates less wave energy partly
due to the stoppage of propagating bore and partly due to the reduction of bottom friction.
As for the effect of reef-crest width (Fig. 11d), although the presence of a reef crest is
reported to be an important factor affecting the wind wave transformation over fringing
reefs (e.g., Yao et al., 2017), it seems to have little impact on the solitary wave runup
under both $h_r$ , slight decline of $R$ could only be found under the crest width larger than
0.4. This is because the solitary wave is very long compared to the reef-crest width, thus
most of its energy could transmit over the narrow reef crest. However, when the reef crest



becomes sufficient wide, its shallower crest tends to energize the wave breaking thus the
energy dissipation. To summarize all above analyses, it can be concluded that coastal
regions protected by the fringing reefs with steeper back-reef slopes and wider lagoons
are more valuable to coastal inundation during a tsunami event.

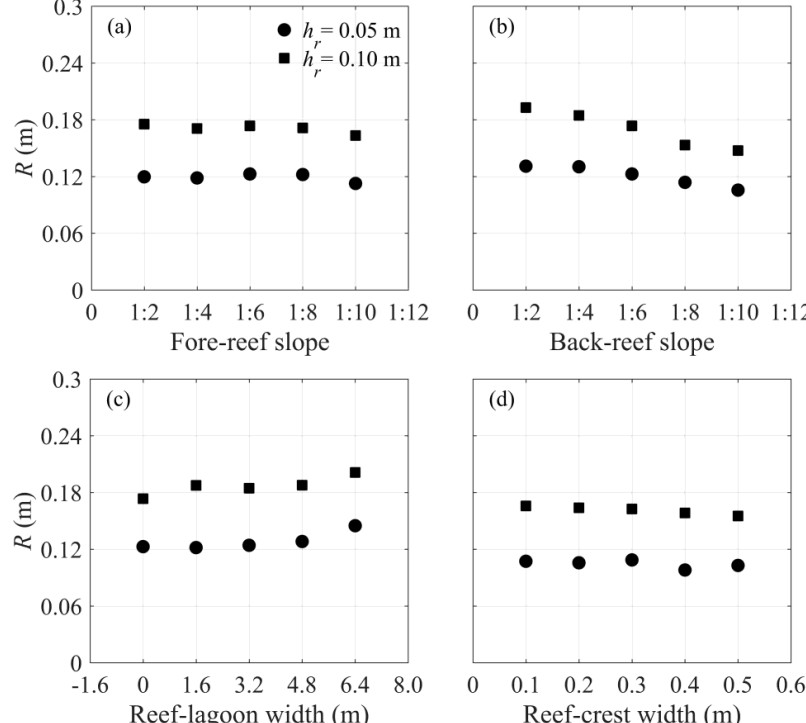


Fig. 11 The predicted wave runup on the back-reef beach ($R$) with the reef morphology
variations under $H_0 = 0.08$ m for different: (a) fore-reef slopes; (b) back-reef slopes; (c)
lagoon widths; and (d) reef-crest widths.
**4.2 Wave-driven current and vortices around the reef crest and the lagoon**

One advantage of the current Navier-Stokes-equation-based model over the depth-

integrated models is its ability to resolve the vertical flow structure under breaking waves,
particularly around the complex reef geometry. Based on the reef profile of Yao et al.
(2018), Fig. 12 shows the simulated wave-driven current and vorticity on the x-z plane at
different stages ($t / \sqrt{h_0 / g}$) for the reefs with and without a reef crest at the reef edge
subjected to the same solitary wave condition ($H_0 = 0.08$ m and $h_r = 0.05$ m). Without




the reef crest, shoaling wave propagates onto the horizontal reef flat with a uniform
velocity distribution underneath ($t/\sqrt{h_0/g} = 25.9$ and $26.9$), which is typical for the
shallow-water long waves. Until to $t/\sqrt{h_0/g} = 27.9$, wave breaking occurs in the form
of a plunging breaker, and vortex generation gathers mainly around the wave crest. The
vortices are transported further downstream at $t/\sqrt{h_0/g} = 28.9$. When the wave crest
exists, incipient wave breaking moves seaward and it takes place at the seaside edge of
the reef crest ($t/\sqrt{h_0/g} = 25.9$). The breaker then overtops over the reef crest
($t/\sqrt{h_0/g} = 26.9$) and plunges onto the reef flat leeside of the reef crest, resulting a
hydraulic jump ($t/\sqrt{h_0/g} = 27.9$). Consequently, wave-driven current at the rear part of
the reef crest is dramatically increased compared to the same location without the crest.
Both the intensity and the extent of vortex generation are also enlarged at the leeside of
the reef crest ($t/\sqrt{h_0/g} = 28.9$), leading to increased wave energy dissipation compared
to the case without the reef crest.

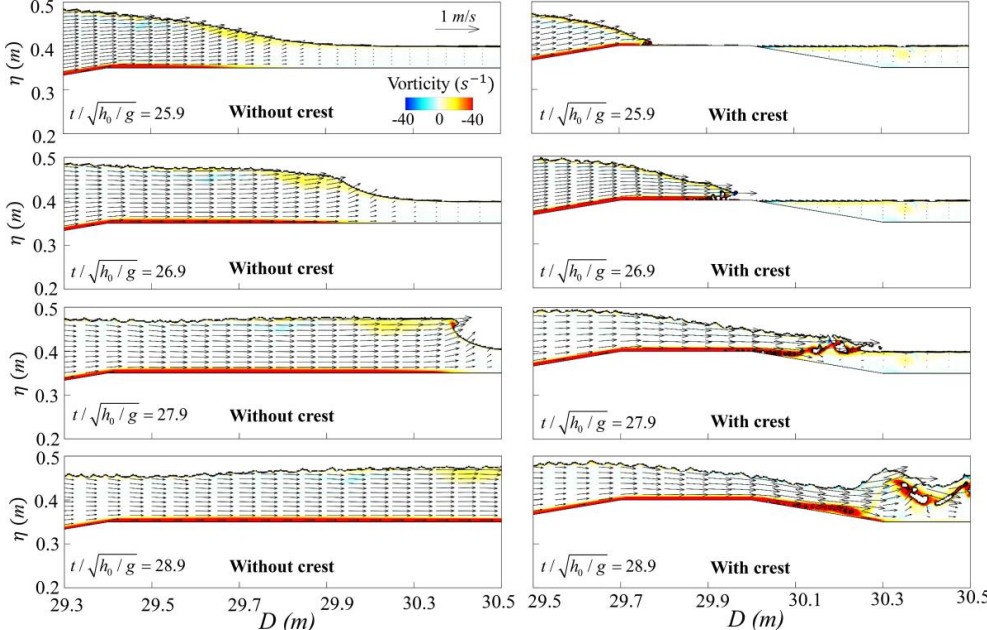


Fig. 12 Comparison of wave-driven current and vorticity on the x-z plane at different
stages ($t/\sqrt{h_0/g}$) between the reefs with and without the reef crest ($H_0 = 0.08$ m and
$h_r = 0.05$ m).



Fig. 13 compares the computed wave-driven current and vorticity on the x-z plane
at different stages ( $t/\sqrt{h_0/g}$ ) between the reefs in the presence and absence of the
lagoon. Without the lagoon, the propagating bore arrives with strong vortex motions
( $t/\sqrt{h_0/g}$ =49.4 ). The vortices are eventfully transported downstream from
$t/\sqrt{h_0/g}$ = 54.4 to 64.4. However, when the lagoon is present, the current speed over
the depth slows down and additional vortices generate at the seaside edge of the lagoon as
the bore propagates into the lagoon ( $t/\sqrt{h_0/g}$ =49.4). The peak value of the vorticity
appears at a later time ( $t/\sqrt{h_0/g}$ = 54.4 ). After that, the vortices in the lagoon are
primarily diffused by the vortex shedding ( $t/\sqrt{h_0/g}$ = 59.4 and 64.4 ). Compared to the
case without the lagoon, although the existence of a lagoon dissipates less wave energy
by terminating the propagating bore and reducing the reef-flat friction as previously
stated, the vortex generation and diffusion in the lagoon as demonstrated here is believed
to cause local energy loss. We finally remark that the wave-driven current and vortices
examined in this section could provide a first step to analyze more sophisticated problems,
such as the tsunami-induced sediments/debris transport over the fringing reefs.

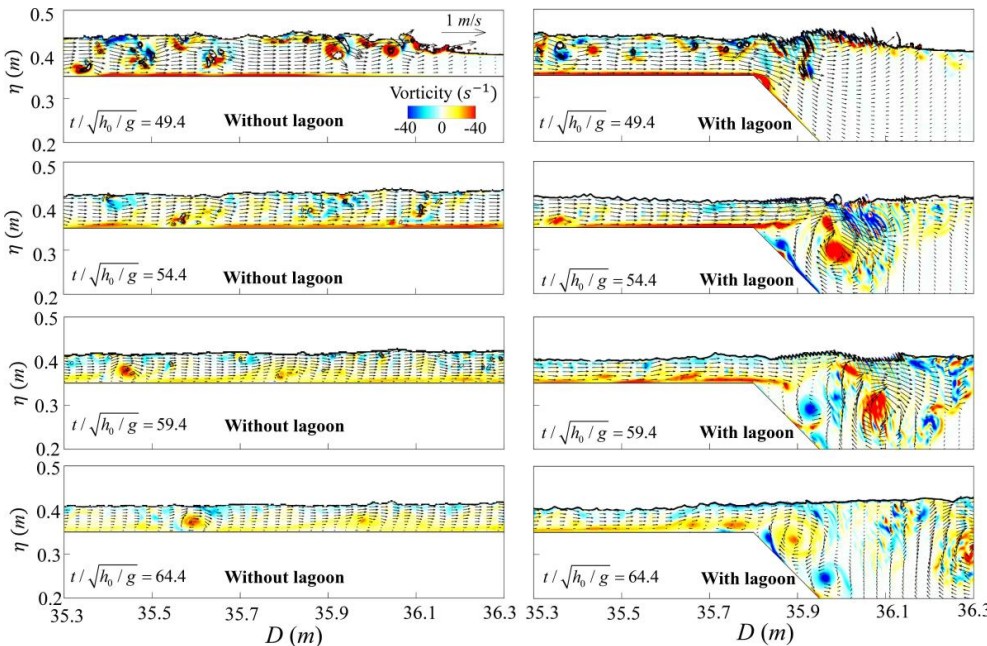




Fig. 13 Comparison of wave-driven current and vorticity on the x-z plane at different
stages ( $t/\sqrt{h_0/g}$ ) between the reefs with and without the lagoon ( $H_0 = 0.08$ m and
$h_r = 0.05$ m).

**5 Conclusions**

To remedy the inadequacies of using the depth-integrated models to simulate the
interaction between tsunami-like solitary waves and fringing reefs, a 3D numerical wave
tank, solving the Navier-Stokes equations with the LES for turbulence closure, has been
developed based on the open-source CFD tool OpenFOAM®. The free surface is tracked
by the VOF method. Two existing laboratory experiments with the wave, flow and wave
runup measurements based on different fringing reef profiles are employed to validate the
numerical model. Simulations show that the current Navier-Stokes-equation-based model
outperforms the commonly used Boussinesq-typed models in view of its capability to
better reproduce the breaking waves and wave-driven current on the reef flat. The model
is then applied to investigate the impacts of varying morphologic features on the back-
reef wave runup. The flow and vorticity fields associated with the breaking solitary wave
around the reef crest and the lagoon are also analyzed via the numerical simulations.
Model results shows that wave runup on the back-reef slope is most sensitive to the
variation of the back-reef slope, less sensitive to the lagoon width, and almost insensitive
to the variations of both the fore-reef slope and the reef-crest width within our tested
ranges. The existence of a reef crest or a lagoon can notably alter the wave-driven current
and vortex evolutions on the reef flat. These findings demonstrate that low-lying coastal
areas fringed by coral reefs with steep back-reef slopes and larger lagoons are expected to
experience larger wave runup near the shoreline, thus they are more susceptible to the
coastal inundation during a tsunami event.

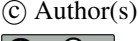




## Acknowledgements

This study was supported financially by the National Natural Science Foundation of
China (grant nos.51679014 and 11702244), the Hunan Science and Technology Plan
Program (Grant No. 2017RS3035), and the Open Foundation of Key Laboratory of
Coastal Disasters and Defense of Ministry of Education (grant no.201602).





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
