# Peer review of "LES Modeling of Tsunami-like Solitary Wave Processes over Fringing Reefs"

_Natural Hazards and Earth System Sciences, 2018_

## Referee Comment (RC1) · Anonymous Referee #1 · 2 Mar 2019

1. General Comments Extreme meteo-ocean phenomena are the most destructive in nature. Tsunami waves are a major component of this threat. Therefore, the subject addressed by the authors is of interest to the readers of NHESS. The authors managed to explain relatively well the methodology - 3D numerical modeling, validated with laboratory data - and applications via a sensitive study (not very extensive) of some environmental characteristics. They use an open source numerical model and claim that the filtered Navier-Stokes (so primitive equations), with a LES approach to close the equations, will be better in comparison to vertical integrated ones (e.g. Boussinesq). The title is appropriate, the conclusions are in line with the sensitive numerical experiments (although restricted by the relatively small range of variation of the parameters) and the references are useful. There is no major issue with the English language and

so the text is understandable. There is not much novelty in this paper but the contribution to the advancement of the knowledge in the behaviour of the fringing reefs is important since these coastal morphologies are common in areas prone to tsunami activity. 2. Specific Comments Point 3.1 describe the experimental settings. These settings correspond to what Froude number? (See your own remarks about the reef-crest widths in page 21 and the implicit scale). Point 3.2 describe the numerical settings. An estimation of the turbulence scales is missing (or the Reynolds number). This way the choice of the discretization would be better understood and justified. With LES this relationship is crucial. Also with LES the imported turbulence at the boundary is usually need to feed the equations, in particular the coherent macro structures to be directly modeled. Some information is missing. Point 3.3 describe the experimental observations and the simulations. The authors refer that some of the discrepancies could be due to "... air entrainment in measuring ...". Actually, what are you comparing ? LES equations are a space filtered non-stationary approach. What are the statistical tools used to tackle both the experimental data and the numerical simulations? 3. Technical corrections. Page 3 (lines 55 and 57) - I think that "scholars" could be replaced with "researchers". Page 4 (line 70) - This is an incomplete description. Mean flows ′models are not phase-averaged. Page 4 (line 71) - Please include also the vortex-force concept and the references. Page 6 (line 149) - The rho is missing in equation (3). Page 6 (line 150) - The rho and miu should be defined after equation (4) and the numerical values given after (9). Page 6 (line 155) - It is not the SGS stress, it is the residual stress. Page 7 (lines 168 and 169) - You are not solving both phases. There are not equations for the air. VOF is not an hydrodynamical equation for the air. It is an equation for the position of the free surface (the two fluids are decoupled). Page 11 (line 256) - The vertical discretization is not kept constant. A better explanation is needed and should be done by direction. Page 14 (line 327) -The effect of the air-bubbles. Any guess of what is the magnitude of this effect? On the other hand, doesn't this influence also the performance of Roeber (2010)? And at D=61,6 m this one is better (at the transition from subcritical to supercritical). Page 21 (line 449) - It is not clear what is the "growth

of the back-beach slope"? 1:12 is larger than 1:2 ? There are two references not in the text: Roeber et al. (2010) and Titov et al. (2005).

---

## Author Comment (AC1) · 21 Mar 2019

Please see the supplementary .zip file in which both the response to the referee #1 and the revised manuscript are included.

Please also note the supplement to this comment:
https://www.nat-hazards-earth-syst-sci-discuss.net/nhess-2018-376/nhess-2018-376-AC1-supplement.zip

---

## Referee Comment (RC2) · Anonymous Referee #2 · 2 May 2019

General comments:

The authors used OpenFOAM to study effects of the key characteristics of fringing-reef profiles on the runup on back-reef beach. The numerical model was validated by comparison with two existing sets of experimental results. The manuscript is acceptable after a minor revision.

Specific comments:

(1) What is Delta-bar in Eq. (7)? (2) What does the index i represent in Eq. (10)? (3) The caption of Figure 11: "for different" is grammatically incomplete. (4) Are there any measured runup data that can be used for model validation? (5) Will the depth-integrated equations over-predict or under-predict the runups?

---

## Author Comment (AC2) · 11 May 2019

Please see the supplement.zip file in which both the responses to the Referees #1 and #2 and the revised manuscript are included.

Please also note the supplement to this comment:
https://www.nat-hazards-earth-syst-sci-discuss.net/nhess-2018-376/nhess-2018-376-AC2-supplement.zip

---

## Author Response (AR1)

**Response to Reviewer 1#**

The authors thank the reviewer for the valuable comments. In the following, we reply the reviewer's comments item-by-item. The reviewer's comments are in *italic*. The corresponding revisions in the revised manuscript are highlighted by the red-color texts.

**1. General Comments**

*Extreme meteo-ocean phenomena are the most destructive in nature. Tsunami waves are a major component of this threat. Therefore, the subject addressed by the authors is of interest to the readers of NHESS. The authors managed to explain relatively well the methodology - 3D numerical modeling, validated with laboratory data - and applications via a sensitive study (not very extensive) of some environmental characteristics. They use an open source numerical model and claim that the filtered Navier-Stokes (so primitive equations), with a LES approach to close the equations, will be better in comparison to vertical integrated ones (e.g. Boussinesq). The title is appropriate, the conclusions are in line with the sensitive numerical experiments (although restricted by the relatively small range of variation of the parameters) and the references are useful. There is no major issue with the English language and so the text is understandable. There is not much novelty in this paper but the contribution to the advancement of the knowledge in the behaviour of the fringing reefs is important since these coastal morphologies are common in areas prone to tsunami activity.*

**2. Specific Comments**

*(1). Point 3.1 describe the experimental settings. These settings correspond to what Froude number? (See your own remarks about the reef-crest widths in page 21 and the implicit scale).*

**Reply:** Since this is a numerical study, we only briefly provided a description of the physical model design as well as the instrument settings in Section 3.1. Description of the prototype reefs can be found in the references, i.e., Roeber (2010) and Yao et al. (2018). More details about the tested wave conditions and the model configurations are given by both Figure 1 and Table 1. To address the reviewer's concern, we have added the following descriptions starting from Line 235 in the revised manuscript.
"The dimensions of the fore-reef slope and the reef flat, the water depths over the reef flat, and the incoming wave heights were designed according to the Froude similarity with a target geometric scale factor of 1:20."

Please note that Roeber (2010) did not give a specific value of the model scale, and He only mentioned that it was a large-scale experiment.

*(2). Point 3.2 describe the numerical settings. An estimation of the turbulence scales is missing (or the Reynolds number). This way the choice of the discretization would be*

*better understood and justified. With LES this relationship is crucial. Also with LES the imported turbulence at the boundary is usually need to feed the equations, in particular the coherent macro structures to be directly modeled. Some information is missing.*

**Reply:** We have added the following descriptions in the revised manuscript.

(1) Line 256:
When solitary waves interact with the investigated laboratory reefs, strong turbulence is expected to be generated inside the domain where wave breaks near the reef crest and propagates on the reef flat as a moving bore, thus we do not set the inflow boundary condition with desired turbulence characteristics for the LES at the wave generation boundary. Meanwhile, since both the laboratory reef surfaces are very smooth, the flow structure near the bottom is not resolved in our simulations, and we only impose a no-slip boundary condition at the reef surfaces by adjusting the velocity near the bottom to satisfy the logarithmic law of the wall.

(2) Line 286:
    For LES modelling solitary wave breaking over reefs, it is crucial to examine the Reynolds number ( Re ) at the incipient breaking point where strong turbulence is generated. It could be calculated by $\text{Re} = u_b\left(H_b + h_b\right)/\nu$ with $u_b = c_b H_b / h_b$ and $c_b = \sqrt{g\left(H_b + h_b\right)}$ , where $H_b$ , $h_b$ , $u_b$ and $c_b$ are wave height, water depth, streamwise velocity and wave celerity at the breaking point, respectively. Re is estimated for all tested scenarios by using $H_b = H_0$ and $h_b = h_r$ (i.e., ignoring wave shoaling on the fore-reef slope and assuming wave breaking at the reef edge) and the values are also given in Table 1. Since the near-wall eddies are not resolved in this study, the total required grid number is independent of Re (Pope, 2000). Ideal grid size of the LES model should be down to the Kolmogorov scale which is infeasible due to the limitation of computational resources. To test the convergence of grid size, we take the experiment with smaller wave flume (i.e., Scenario 3 in Table 1) which requires finer grid resolution as an example. We only examine the grid across the reef profile (the aforementioned core region) where the effect of grid size is supposed to be most influential. Both grid sizes ( $\Delta x$ and $\Delta z$ ) ranging from 8 mm down to 1 mm are tested. The results in terms of the dimensionless free surface elevation and streamwise velocity associated with the leading solitary wave in the inner reef flat (G7) are compared in Fig. 3. Only less than 2% differences in terms of wave and flow could be observed with the grid size varying from 2 mm to 1 mm, indicating that our selection of grid size $\Delta x = \Delta z = 2$ mm is sufficient for the current simulations.

[Figure]

**Fig. 3.** Variation of the maximum dimensionless free surface elevation ($\eta_{max}/H_0 > 1$) and streamwise velocity ($u_{max}/\sqrt{gh} > 1$) at G7 with the grid size ($\Delta x$ and $\Delta z$) across the reef for Case 3 in Table 1.

Table 1 Reef configuration and wave condition for the tested scenarios

| Scenario I.D. | Offshore wave height $H_0$ (m) | Offshore water depth $h_0$ (m) | Reef-flat water depth $h_r$ (m) | Fore-reef slope $s$ | Reef-flat length $L_r$ (m) | Reynolds number Re at the breaking point | Remarks | Source |
|---|---|---|---|---|---|---|---|---|
| 1 | 1.23 | 2.46 | 0.1 | 1:12 | 29.5 | $5.9 \times 10^7$ | − | Roeber (2010) |
| 2 | 0.75 | 2.5 | 0.14 | 1:12 | 22.8 | $1.4 \times 10^7$ | With reef crest | Roeber (2010) |
| 3 | 0.08 | 0.40 | 0.05 | 1:6 | 9.6 | $2.4 \times 10^5$ | − | Yao et al. (2018) |
| 4 | 0.08 | 0.40 | 0.05 | 1:6 | 8.0 | $2.4 \times 10^5$ | With lagoon | Yao et al. (2018) |

*(3). Point 3.3 describe the experimental observations and the simulations. The authors refer that some of the discrepancies could be due to "... air entrainment in measuring ...". Actually, what are you comparing ? LES equations are a space filtered non-stationary approach. What are the statistical tools used to tackle both the experimental data and the numerical simulations?*

**Reply:** We are comprising the time series of free surface elevation and streamwise velocity at specific cross-reef locations, including those in the surf zone. For the laboratory data, they are the measured time series from the instruments, while in the numerical simulations, they are obtained from the computed time series of filtered

pressure and velocity. Since the time series consist of the instantaneous free surface elevation/velocity at each time step, no statistical tools are used in our data analysis.

For the experiments, Roeber (2010) has also claimed that the instrument accuracy may be affected by the air bubbles in the surf zone, and we think it is difficult to quantify this measurement error. For the simulations, the error may come from the difficulty in tracking the true free surface where the volume fraction of water, i.e., $\alpha$ in Eq. (9) is set to 0 for the air bubbles. Therefore, for the completeness, we have updated our descriptions in the manuscript as follows:

(1) Line 345
Such discrepancies may be primarily due to the air entrainment in measuring both the breaking wave and the moving bore (Roeber, 2010) as well as the air bubble effect on free surface tracking by the VOF method.

(2) Line 364
"and less satisfactory prediction (skill values =0.76) at this location is probably again due to the effect of air-bubbles on both flow measurements in the experiments and free surface tracking in the simulations."

**3. Technical corrections**

*(5). Page 3 lines 55-57. I think that "scholars" could be replaced with "researchers".*
**Reply:** We have replaced "scholars "by "researchers" at Line 57 in the revised manuscript.

*(6). Page 4 line 70. This is an incomplete description. Mean flows 0models are not phase-averaged.*
**Reply:** We have updated the description starting from Line 69 in the revised manuscript, which is also shown below "The first group focuses on using the phase-averaged wave models and the nonlinear shallow water equations to model the waves and the flows, respectively, in field reef environments."

*(7). Page 4 line 71. Please include also the vortex-force concept and the references.*
**Reply:** We have updated the description starting from Line 71 in the revised manuscript, which is also shown below "typically the concept of radiation stress (Longuet-Higgins and Stewart, 1964) or vortex-force (Craik and Leibovich, 1976) is used to couple the waves and the flows."
The following reference has been added in the reference list:
Craik, A. D., and Leibovich, S.: A rational model for Langmuir circulations. J. Fluid Mech, 73(3), 401–426, 1976.

*(8). Page 6 line 149. The rho is missing in equation (3).*
**Reply:** We have added a description at Line 149 in the revised manuscript, which is

also shown below: " For incompressible flow, ….."

*(9). Page 6 line 150. The rho and miu should be defined after equation (4) and the numerical values given after (9).*

**Reply:** We have updated the description starting from Line 153 in the revised manuscript, which is also shown below: "where $\rho$ is the water density, $\mu$ is the dynamic viscosity," We have already defined $\rho_1$ and $\rho_2$ after Equation (9).

*(10). Page 6 line 155. It is not the SGS stress, it is the residual stress.*

**Reply:** We have replaced "SGS" by "residual" at Line 158 in the revised manuscript.

*(11). Page 7 lines 168-169. You are not solving both phases. There are not equations for the air. VOF is not an hydrodynamical equation for the air. It is an equation for the position of the free surface (the two fluids are decoupled).*

**Reply:** We have updated the description starting from Line 171 in the revised manuscript, which is also shown below: "The presence of the free-surface interface between the air and water is treated through the commonly used VOF method (Hirt and Nichols, 1981), which introduces a volume fraction and solves an additional modeled transport equation for this quantity."

*(12). Page 11 line 256. The vertical discretization is not kept constant. A better explanation is needed and should be done by direction.*

**Reply:** We have updated the description starting from Line 273 in the revised manuscript, which is also shown below "For the vertical ( $z$ ) direction, the grid size is initially set to be $\Delta z$= 20 mm/8 mm across the domain for Roeber/Yao et al.'s scenarios. Grid refinement near the free surface (e.g., Figs. 2b and 2c) is also conducted across the domain by reducing the grid sizes to $\Delta z$= 5 mm/2 mm."

*(13). Page 14 line 327. The effect of the air-bubbles. Any guess of what is the magnitude of this effect? On the other hand, doesn't this influence also the performance of Roeber (2010)? And at D=61,6 m this one is better (at the transition from subcritical to supercritical).*

**Reply:** Please see our reply of the Specific Comments - Point 3.3 above.

*(14). Page 21 line 449. It is not clear what is the "growth of the back-beach slope"? 1:12 is larger than 1:2 ?*

**Reply:** We have updated the description starting from Line 486 in the revised manuscript, which is also shown below: "It decreases significantly as the back-beach slope becomes milder,"

*(15) There are two references not in the text: Roeber et al. (2010) and Titov et al. (2005).*

**Reply:** We have deleted the two excessive references in the reference list.

**Response to Reviewer 2#**

The authors thank the reviewer for the valuable comments. In the following, we reply the reviewer's comments item-by-item. The reviewer's comments are in *italic*. The corresponding revisions in the revised manuscript are highlighted by the red-color texts.

**1. General Comments**

*The authors used OpenFOAM to study effects of the key characteristics of fringing-reef profiles on the runup on back-reef beach. The numerical model was validated by comparison with two existing sets of experimental results. The manuscript is acceptable after a minor revision.*

**2. Specific Comments**

*(1) What is Delta-bar in Eq. (7)?*
**Reply:** Sorry for the typo, it should be Delta, which has been given by Eq. (2). We have corrected it in the revised manuscript.

*(2) What does the index i represent in Eq. (10)?*
**Reply:** We have added the following description at Line 140 in the revised manuscript for the subscript i when it appears at the first time in the manuscript, i.e., in Eq. (1): " The filtered velocity in the i-th spatial coordinate is defined as  ".

*(3). The caption of Figure 11: "for different" is grammatically incomplete.*
**Reply:** We have revised the figure caption (Figure 11 in the original manuscript, Figure 12 in the revised manuscript) as follows:
Fig. 12 The predicted wave runup on the back-reef beach ( $R$ ) under $H_0 = 0.08$ m with varying: (a) fore-reef slopes; (b) back-reef slopes; (c) lagoon widths; and (d) reef-crest widths.

*(4). Are there any measured runup data that can be used for model validation?*
**Reply:** Since the laboratory experiments reported by Yao et al.(2018) only measured the maximum wave runup (not the time series of shoreline movement), we have in fact described the measured and predicted runups for Scenarios 3 and 4 in the texts, such descriptions can be found at:
(1) Line 435
The predicted and measured wave runups are 0.122 m and 0.109 m, respectively, for this scenario. Compared to the Boussinesq model employed by Yao et al. (2018), no significant difference in the predicted timeseries could be found for the present Navier-Stokes-equation-based model.
(2) Line 446

Again, the model performance for this scenario is fairly good (all skill values larger than 0.9). The predicted and measured wave runups are 0.123 m and 0.116 m, respectively, for this scenario.

*(5). Will the depth-integrated equations over-predict or under-predict the runups?*
**Reply:** The model over-predicted the runups for both scenarios, please see also the reply of Comment 4 above.